# Simultaneous Removal of Polymers with Different Ionic Character from Their Mixed Solutions Using Herb-Based Biochars and Activated Carbons

**DOI:** 10.3390/molecules27217557

**Published:** 2022-11-04

**Authors:** Marlena Gęca, Małgorzata Wiśniewska, Piotr Nowicki

**Affiliations:** 1Department of Radiochemistry and Environmental Chemistry, Faculty of Chemistry, Institute of Chemical Sciences, Maria Curie- Sklodowska University in Lublin, M. Curie-Sklodowska Sq. 3, 20-031 Lublin, Poland; 2Department of Applied Chemistry, Faculty of Chemistry, Adam Mickiewicz University in Poznań, Uniwersytetu Poznańskiego 8, 61-614 Poznań, Poland

**Keywords:** activated carbon, biochar, simultaneous polymers adsorption, poly(acrylic acid), polyethylenimine, organic substances removal

## Abstract

Nettle and the sage herbs were used to obtain carbonaceous adsorbents. For the biochar preparation the precursors were dried and subjected to conventional pyrolysis. Activated carbons were obtained during precursor impregnation with phosphoric(V) acid and multistep pyrolysis. The textural parameters and acidic-basic properties of the obtained adsorbents were studied. The activated carbons prepared from the above herbs were characterized by the largely developed specific surface area. The obtained carbonaceous adsorbents were used for polymer removal from aqueous solution. Poly(acrylic acid) (PAA) and polyethylenimine (PEI) were chosen, due to their frequent presence in wastewater resulting from their extensive usage in many industrial fields. The influence of polymers on the electrokinetic properties of activated carbon were considered. PAA adsorption caused a decrease in the zeta potential and the surface charge density, whereas PEI increased these values. The activated carbons and biochars were used as polymer adsorbents from their single and binary solutions. Both polymers showed the greatest adsorption at pH 3. Poly (acrylic acid) had no significant effect on the polyethylenimine adsorbed amount, whereas PEI presence decreased the amount of PAA adsorption. Both polymers could be successfully desorbed from the activated carbons and biochar surfaces. The presented studies are innovatory and greatly required for the development of new environment protection procedures.

## 1. Introduction

Limited reserves of natural resources are one of the most important problems facing present and future generations. Therefore, it is necessary to find new, nontoxic and renewable resources. Various types of green plants, capable of rebuilding the harvested parts of shoots, could be the solution to this global problem. Annual plants, such as herbs, grow and decompose through the year so their harvesting has no negative influence on the natural environment. The judicious use of raw plant materials would enable them to be an inexhaustible resource for different fields of industry [1].

Biochar and activated carbons prepared from plant material are environmentally friendly solids with largely developed surface areas and extensive porous structures. They are relatively easy to obtain, low-cost and regenerable adsorbents. Biochar and activated carbons are often used as heavy metal adsorbents [2,3,4]. Organic compound adsorption on the surface of these solids has been extensively studied [5,6]. However, the most common research has been concerned with the removal of dyes and pharmaceuticals from the liquid phase [7].

The usage of different kinds of polymers in modern industry and everyday life is constantly expanding. Poly(acrylic acid) (PAA) is used as a suspension stabilizer or destabilizer, as well as a dispersant agent in cosmetics and drugs [8,9,10]. In turn, polyethylenimine (PEI) finds application as the additive for papermaking and cell line cultures [9,11]. Both of the above polymers find application as a complexing agent for the removal of heavy metal ions from water [12]. The increasing demand for polymers has increased their amounts in sewage. Despite their wide usage, these macromolecular compounds can be dangerous for the environment. Their presence can disturb the natural balance and have negative effects on all living organisms. One of the ways to prevent polymeric pollution is their adsorption on activated carbons and biochars. Polymers adsorption on the surface of carbonaceous adsorbents has rarely been described. However, recent scientific reports have proved that the application of activated carbons or biochar for the removal of polymeric substances is not only possible, but also effective [13,14].

In the present study biochar and activated carbons, obtained from nettle and sage herbs, were used as poly (acrylic acid) and polyethylenimine adsorbents from their single and binary solutions. The textural parameters and acidic–basic properties of the obtained materials were determined. The influence of the presence of poly (acrylic acid) and polyethylenimine in the system on surface and electrokinetic properties (connected with the structure of the electrical double layer) of carbon sorbents was also studied. The effect of solution pH on the adsorption process was evaluated and its mechanism proposed. Additionally, the regeneration possibilities of the applied materials were examined and the most effective desorbing agents were selected. As mentioned above, there has been little research on the usage of biochars or activated carbons as macromolecular substance adsorbents. The presented results are even more innovative because the simultaneous removal of two polymers with different ionic characters using biochar or activated carbons has not been described before. The description of the structure of the mixed polymeric layer enables understanding of the mechanisms governing the binding of macromolecules at the interface, and, thus, of the optimization of procedures for their removal from the liquid phase during wastewater treatment.

## 2. Results

### 2.1. Physicochemical Properties of the Prepared Adsorbents

Table 1 shows the elemental composition of the obtained adsorbents. All solids contain mainly carbon (78–90 wt. %); however, the products of chemical activation with H_3_PO_4_ contain significantly more of this element. The obtained materials also differed, quite significantly, with regard to nitrogen content. For pyrolysis products, the content of N^daf^ (^daf^—dry-ash free basis) ranged from 2.77 to 1.78 wt. %, while in the analogous activated carbons it was clearly lower and reached 1.13 wt. % for the NE_AC sample obtained from the nettle herb and 0.24% for the SA_AC biochar obtained from the sage herb. Mg, Ca and K were present only in the biochar structure, and the activated carbons did not contain these elements, which were probably removed during the activation process. As indicated by the SEM pictures presented in Figure 1, the obtained carbonaceous materials also exhibited some differences in morphology.

However, much greater differentiation was observed in the case of textural parameters. As follows from the data collected in Table 2, both activated carbons had a largely developed surface area of over 800 m^2^/g; however, the product derived from the sage herb fared better in this regard. Both biochars surface areas were slightly over 2 m^2^/g, which proved that the activation process had a much greater influence on the porous structure formation of the absorbents. Activated carbon pore volumes were also much greater than those obtained for the biochars. The mean pore size was about 4 nm for the activated carbons and about 10 nm for the biochar, which allowed classification of all adsorbents as mesoporous materials. Larger pore sizes are desirable for the adsorption of large polymer macromolecules. The yields of the pyrolysis and activation processes were below 50% and higher for the adsorbents obtained from the sage herb. The total amount of the surface functional group (Table 3) was greater for the biochars in comparison to the activated carbons. The NE_B biochar was characterized by a significantly greater amount of basic surface groups than the acidic ones, whereas the SA_B sample contained comparable contents of both types of functional groups. In turn, the activation products (especially the NE_AC sample) were characterized by a significant predominance of acidic groups on their surfaces. What is more, a larger total amount of surface functional groups was observed in the case of the solids obtained from the nettle herb. 

### 2.2. Adsorbents Electrokinetic Characteristic

The studies on the activated carbon surface charge densities confirmed the acidic properties of both adsorbents. The point of zero charge (pzc) is a pH value at which the concentration of positively charged surface groups is the same as the concentration of negatively charged ones, so the total surface charge is equal to zero. The pzc of the NE_AC activated carbon occurred at pH 3.1 and in the case of SA_AC at pH 4.0 (Figure 2). The poly (acrylic acid) addition caused a slight decrease of the σ_0_ value in the whole range of studied pH in comparison to the system without the polymer. This was caused by the presence of negatively charged carboxyl groups of the adsorbed PAA macromolecules in the by-surface layer of the solution. This effect has been previously described [15,16]. On the other hand, the polyethylenimine presence in the solution caused increase in the surface charge density. This resulted from the presence of the cationic amino groups of adsorbed PEI macromolecules in the by-surface layer of the solution, which were not directly bound with the solid surface, similar to PAA interfacial behavior [17]. In the case of the positively charged PEI chains, its conformation on the negatively charged surface (above pH_pzc_, namely 3-4) was flatter (in comparison to the negatively charged PAA chains), which assured a large accumulation of positive charges in the surface layer.

The pH_pzc_ value for the NE_AC activated carbon in the presence of both polymers was 4.0, and for the SA_AC, 4.1. This was an intermediate effect compared to the single polymer systems. Positively charged polyethyleneimine and negatively charged poly (acrylic acid) were both adsorbed onto the activated carbons at the same time, which caused the pH_pzc_ value in the binary solution to be between the pH_pzc_ value of the single PEI solution and the single PAA solution. 

Both applied biochars were characterized by basic properties of the surface. The NE_B biochar pH_pzc_ occurred at 7.3 and for the SA_B sample at 8.6. The difference in the acid/base properties of the biochars and activated carbons was evidently associated with the use of H_3_PO_4_ as the activating agent.

The zeta potential studies showed that the isoelectric point (iep) occurred at pH 4.9 for the NE_AC activated carbon and at pH 6.2 for the SA_AC one. The pH_iep_ value for the NE_B biochar occurred at 6.1 and for the SA_B one at 5.2 (Figure 3). The isoelectric point is a pH value at which the number of adsorbate ions/functional groups with positive and negative charges, accumulated in the slipping plane area, are the same (the zeta potential has a zero value). For all materials there was a difference between the pH_pzc_ and pH_iep_ values. The activated carbons isoelectric point occurred at higher pH values than the point of zero charge of these adsorbents and for the biochar the pH_pzc_ value was higher than pH_iep_. This was related to the partial overlapping of the electrical double layers (edl) that formed on the pore walls, the effects of which on mesoporous materials have been previously described in the literature [14,18]. The poly (acrylic acid) presence in the suspension caused a decrease in the zeta potential value. The opposite tendency was observed after the addition of polyethyleneimine. This could be explained by the presence of the ionic polymer groups (negatively charged in the case of PAA and positively charged for PEI) in the slipping plane area [19]. The zeta potential value in the suspension containing both polymers increased in comparison to the suspension without any adsorbates, which could have been caused by a few different factors, e.g., the presence of molecules with the opposite charge as well as the shift of the slip plane [20]. The negatively charged PAA macromolecules were repulsed from the negatively charged activated carbon surfaces at higher pH values which caused shift of the slipping plane by the polymeric chains adsorbed perpendicularly to the solid surface and the pH_iep_ value decreased. In turn, the presence of the positively charged PEI molecule (which was adsorbed flatter) in the slipping plane area caused the pH_iep_ value to increase. Generally speaking, the simultaneous presence of both polymers on the adsorbent surface had an intermediate effect, in comparison to the single polymer systems.

Table 4 presents the activated carbons and biochar aggregates sizes. The obtained data proved that the presence of the polymers (in particular PAA) had a very small effect on the NE_AC activated carbon aggregate size at pH 3. The aggregate sizes in the single adsorbate systems were also similar to the system without any adsorbate at pH 6 and 9. However, both polymers being present in the solution could form polymer–polymer complexes at higher pH values, which resulted in the larger size of NE_AC activated carbon aggregates in the binary solution at pH 6 and 9. The presence of poly (acrylic acid) considerably affected the SA_AC activated carbon aggregate size at pH 3 and 9. On the other hand, at pH 6 the polyethylenimine caused increase in size of the aggregates formed by the activated carbon obtained from the sage herb. Both polymers present in the solution had the greatest effect on aggregate size at pH 6, due to the polymer–polymer complex formations. In general, sizes of aggregates formed by both biochar were considerably larger than those formed by the analogous activated carbons.

### 2.3. Adsorption–Desorption Studies

Figure 4 shows the adsorption isotherms of poly (acrylic acid) and polyethylenimine at pH 3, 6 and 9. Both activated carbons were characterized by the acidic character of the surface so at low pH values they were negatively charged, which favored electrostatic adsorption of the cationic PEI, and, in a way, limited the anionic PAA adsorption. Nevertheless, in the examined system the adsorbed amounts of PAA were larger than those of PEI. The poly (acrylic acid) adsorption was mostly caused by the chemical interactions and hydrogen bond creation. The greater PAA adsorbed amounts were mostly caused by its coiled conformation at pH 3, in contrast to the developed PEI conformation. 

The obtained results proved that the greatest PAA adsorbed amount occurred at pH 3. This was directly related to its conformation, at pH 3 pol y(acrylic acid) assumed the most coiled conformation due to minimal dissociation of its functional groups. For this reason, the adsorption of PAA coils in the pores was also possible under such pH conditions. As a result, the PAA adsorbed amount was the greatest at pH 3 (dense packing of adsorption layers).

At pH 6 and 9 PAA chains developed, which limited the polymer adsorption in the porous structure of the adsorbent. The PAA adsorption on the negatively charged activated carbon surfaces was possible due to hydrogen bond formations [21].

The pH value had a smaller effect on the polyethyleneimine adsorption. This was related to the high value of PEI pK_b_, which was over 9, so at all studied pH values polyethyleneimine assumed the developed conformation. Similar to PAA, polyethyleneimine showed minimally greater affinity for the solid surface at pH 3. In such a case, its adsorption was mostly caused by favorable electrostatic interactions [22]. At pH 3 polyethyleneimine was completely dissociated (it had a positive charge) and the acidic activated carbons were negatively charged at this pH value. Thus, the adsorbent surface attracted the polymer molecules.

In general, polymers were adsorbed in greater amounts on the NE_AC activated carbon in relation to the SA_AC one. This could have been caused by a greater amount of functional group present on the NE_AC surface in comparison to SA_AC. The sample obtained from the nettle herb contained almost twice as many functional groups as the material prepared from the sage herb.

The maximum amount adsorbed on the NE_AC surface at pH 3 was 273 mg/g for PAA and 156 mg/g for PEI. The poly (acrylic acid) adsorbed amount on the NE_AC activated carbon was greater than on the activated carbon obtained from cherry stones, 25 mg/g, in [23], and on the biocarbon obtained from peanut shells, 50 mg/g, as well as on the carbon adsorbent prepared from corncobs, 80 mg/g, in [24].

The kinetic data showed that the equilibrium state was reached after 60 min in the solution containing poly (acrylic acid) and after 10min in the solution containing polyethylenimine (Figure 5). The shorter time in the solution containing PEI was probably caused by a smaller adsorption amount in comparison to the solution containing PAA. The obtained results of kinetic measurements were fitted to the pseudo first- and pseudo second-order equations (Table 5). The data obtained for both polymers were better described by the pseudo second-order model. This indicated that the adsorption of these compounds on the prepared activated carbons mainly involved chemical interactions.

The amounts of polymers adsorbed from the single and binary systems on the surface of activated carbons and biochars at pH 3 are presented in Figure 6. It is clearly visible that poly (acrylic acid) was adsorbed better on the activated carbons than polyethylenimine. Most probably the polymers’ conformations (PAA being coiled and PEI developed) was responsible for this. Both activated carbons had similar textural properties so both types of macromolecules were adsorbed on their surfaces at a similar level. The polyethyleneimine adsorption was not affected by poly (acrylic acid). Small PAA molecules (with the coiled conformation) were probably adsorbed in the pores, so they did not impede the PEI adsorption on the solid surfaces. However, the developed PEI chains could block the porous structure of activated carbons for the poly (acrylic acid) adsorption, which resulted in decrease in the PAA adsorbed amount, in comparison to its adsorption from the single solution. The polymer–polymer complex formations could also affect macromolecule adsorption from the binary solution. However, at pH 3 these complexes hardly formed due to the coiled conformation of undissociated PAA chains.

The polyethylenimine adsorption on the NE_B biochar was difficult to determine due to rinsing out the adsorbent components resulting in increase of the solution’s ionic strength, which made PEI–CuCl_2_ complex formation impossible. In the case of polyethylenimine adsorption on the SA_B biochar, such a problem did not occur. The PEI adsorbed amount was significantly smaller on the biochar (2.17 mg/g) than on the corresponding activated carbon SA_AC (25.46 mg/g). The presence of poly (acrylic acid) had a slightly positive effect on its adsorption (2.87 mg/g). The PAA adsorbed amount (3.67 mg/g, 3.03 mg/g) on the biochar was considerably smaller than on the surface of activated carbons (99.08 mg/g, 89.67 mg/g). The cationic polymer presence in the solution had a positive effect on the poly (acrylic acid) adsorption on the SA_B biochar (3.83 mg/g), but did not practically affect the amount of PAA adsorbed on the NE_B biochar (3.82 mg/g). Smaller adsorption of polymers on the biochars was first of all caused by their unfavorable textural parameters. It should also be noted that poly (acrylic acid) and polyethylenimine could have a positive effect on mutual adsorption due to the polymer–polymer complex formation.

The analysis of the XPS data presented in Table 6 shows that the adsorption of polymer pollutants led to significant changes on the surface of the NA_AC activated carbon. As a result of PAA adsorption, the oxygen content almost doubled, which was most likely due to the formation of a polymeric layer rich in carboxyl groups on the adsorbent surface. On the other hand, adsorption of PEI on the surface of activated carbon was accompanied by an almost threefold increase in nitrogen content, indicating the formation of an adsorbate layer rich in -NH- or -NH_2_ groups. Moreover, the adsorption of polyethyleneimine caused the appearance of small amounts of chlorine in the tested material (the source of which were Cl^-^ supporting electrolyte anions having high affinity to the PEI cationic groups). In the case of the system containing activated carbon and both polymers, there was a significant increase in the content of both nitrogen and oxygen, which indicated a permanent bond of both adsorbates to the adsorbent surface.

The data obtained from polymer desorption studies are presented in Table 7. Poly (acrylic acid) was desorbed using H_2_O, HNO_3_ and NaOH. The greatest desorption effectiveness was obtained for the system NE_AC+PAA using a sodium base (61.01%). Water, as well as nitric acid, were not effective desorption agents (efficacy below 3%). Desorption from the surface of biochars with NaOH was impossible. NaOH rinsed out biochar components which caused the black color of the solution and made spectrophotometric determination impossible. The H_2_O and HNO_3_ turned out to be better PAA desorbing agents from the surface of biochars than from the surface of activated carbons. The greatest poly (acrylic acid) desorption from the biochars structure was obtained for the NE_B + PAA + PEI system using water (34.19%). Due to the increasing ionic strength of the solution, only water was used for the polyethylenimine desorption. The greatest desorption efficiencies were obtained from the systems of NE_AC+PEI (44.15%) and SA_B+PEI (44.08%).

## 3. Materials and Methods

### 3.1. Adsorbates

Poly (acrylic acid), PAA, and polyethylenimine, PEI were applied in this study as macromolecular adsorbates. Poly (acrylic acid) (Fluka, Saint Louis, MO, USA), with an average molecular weight equal to 2000 Da, is a weak polyelectrolyte with anionic character, due to the presence of carboxyl groups in its macromolecules. The PAA pK_a_ value is about 4.5. At this pH value 50% of its carboxyl groups are dissociated [25]. Increase in the dissociation degree of carboxyl groups causes more and more expanded conformation of polymeric chains in the solution. At pH 3 the polymer has the most coiled conformation as the result of minimal PAA ionization.

Polyethylenimine (Sigma Aldrich, Saint Louis, MO, USA), with an average molecular weight equal to 2000 Da, is a cationic polymer. It assumes the highest positive charge density in aqueous solution when it is fully protonated. PEI pK_b_ is about 9 (at this pH value the polyethylenimine dissociation degree is 0.5) and at this pH value, polyethylenimine is in a more coiled form, caused by the partial dissociation of amine groups (the PEI dissociation degree value at pH 6 is about 0.999, whereas at pH 3 it is equal to 1) [26,27]. PEI occurs in the form of branched chains, due to the presence of the amine groups [9].

### 3.2. Biochars and Activated Carbons Preparation

The nettle herb (NE) and the sage herb (SA) were used as precursors of carbonaceous adsorbents. In the first step of biochar preparation, the stalks were cut into 1.5–2.0 cm pieces and dried at 110 °C. The fragmented and dried materials were next subjected to conventional pyrolysis using a horizontal resistance furnace equipped with a quartz tube reactor (one-zone model PRW75/LM, Czylok, Jastrzębie-Zdrój, Poland). About 15 g of the precursors was placed in the nickel boats, heated to 400 °C (with a rate 5 °C/min) and thermostated at the final pyrolysis temperature for 60 min. After pyrolysis, the samples were cooled down in a nitrogen atmosphere (Linde Gaz, Kościan, Poland) with a flow rate of 170 cm^3^/min. The obtained biochars were denoted as NE_B and SA_B. 

In order to obtain the activated carbons, both precursors were impregnated with 50% phosphoric(V) acid solution (STANLAB, Lublin, Poland), at the precursor: activating factor weight ratio equal to 1:2. After 24 h of impregnation process at room temperature, the samples were dried at 110 °C to complete the evaporation of water. Then, the impregnated samples were placed into the quartz boats and heated in a nitrogen atmosphere (flow rate 200 cm^3^/min). In the first stage, the samples were heated to the temperature of 200 °C at a rate of 5 °C/min. Then, the materials were annealed at that temperature for 30 min. In the next step, the samples were heated to the final activation temperature of 500 °C (at the rate of 5 °C/min) and again annealed for 30 min. After that time, the samples were cooled down to room temperature in a nitrogen atmosphere. The solids were washed with 10 dm^3^ of hot distilled water in a vacuum filtration funnel with a glass sintered disc and dried to a constant mass at 110 °C. The obtained activated carbons were denoted as NE_AC and SA_AC. 

### 3.3. Characterization of the Biochars and Activated Carbons

The SEM technique (Quanta 250 FEG by FEI, Waltham, MA, USA) was applied for determination of surface morphology and elemental composition (detector Octane Elect Plus by EDAX, Berwyn, IL, USA) of the examined solids.

The total micropore surface area, total pore/micropore volume and average pore diameter were determined by nitrogen adsorption—desorption measured at −196 °C using ASAP 2420 (Accelerated Surface Area and Porosimetry System) provided by Micromeritics (Norcross, GA, USA).

The surface functional group content was determined according to the Boehm method (back titration method), described in detail in our earlier paper [28]. The volumetric standards of 0.1 mol/dm^3^ sodium hydroxide and 0.1 mol/dm^3^ hydrochloric acid (Avantor Performance Materials, Gliwice, Poland) were used as the titrants. The value of pH was determined by the following procedure: samples of 0.25 g were placed in vials, to each of them 25 cm^3^ of distilled water was added and the content was stirred by a magnetic stirrer for 12 h. The pH of the obtained suspensions was measured by means of a CP –401 pH-meter (ELMETRON, Zabrze, Poland), equipped with a combined glass electrode EPS-1.

The XPS (X-ray photoelectron spectroscopy) apparatus (Gammadata Scienta, Uppsala, Sweden) was used to determine the elemental composition of the surface layer of the NE_AC activated carbon without, and in the presence of, polymeric adsorbates.

### 3.4. Electrokinetic Parameters Determination

The determination of surface charge density (σ_0_) of the biochars and activated carbon particles with or without the selected adsorbates was performed applying the potentiometric titration method. A quantity of 50 cm^3^ of suspension containing 100 ppm of adsorbates, 0.001 mol/dm^3^ of NaCl supporting electrolyte and 0.025 g of NE_AC, 0.025 g of SA_AC or 0.5 g of biochars was used for the purpose. The examined solution was placed in a thermostated Teflon vessel (RE 204 thermostat, Lauda Scientific, Lauda-Königshofen, Germany), in which glass and calomel electrodes (Beckman Instruments, Brea, CA, USA) were introduced to monitor pH changes (pHM 240 pH meter, Radiometer, Warsaw, Poland) after each portion of base—NaOH, with a concentration of 0.1 mol/dm^3^ (automatic Dosimat 765 microburette, Metrohm, Opacz-Kolonia, Poland) was added. The measurements were made at 25 °C. The changes in the σ_0_ value, as a function of solution pH, were calculated with the computer program “Titr_v3”, which also controlled the course of the titration process. These calculations were based on the difference in the base volume added to the suspension and the supporting electrolyte solution providing the specified pH value [29].

The zeta potential (ζ) of the biochars and activated carbon particles with and without selected adsorbates was determined. A quantity of 200 cm^3^ of the suspension containing the polymeric adsorbate with a concentration of 100 ppm, NaCl as the supporting electrolyte (0.001 mol/dm^3^) and 0.03 g of the solid was prepared. These systems were subjected to the action of ultrasounds (XL 2020 ultrasonic head, Misonix, Farmingdale, NY, USA) for 3 min and divided into several parts. In each of the obtained samples a different pH value (changing in the range from 3 to 11) was adjusted. For this purpose, the solutions of HCl and NaOH, with concentrations of 0.1 mol/dm^3^, as well as a Φ360 Ph—meter (Beckman, Brea, CA, USA) were used. The measurements were performed at 25 °C in the suspensions with and without the single and binary adsorbates applying the Doppler laser electrophoresis method and Zetasizer Nano ZS (Malvern Instruments, Malvern, UK): NE_B + NaCl, SA_B + NaCl, NE_AC + NaCl, SA_AC + NaCl, NE_AC + NaCl + PAA, NE_AC + NaCl + PEI, SA_AC + NaCl + PAA, SA_AC + NaCl + PEI, NE_AC + NaCl + PAA + PEI, SA_AC + NaCl + PAA + PEI. The apparatus allowed measurement of the electrophoretic mobility of the solid particles without, and covered with, the adsorbate layers. Based on the obtained results, the zeta potential (ζ) was calculated using Henry’s equation [30]. Additionally, the aggregate sizes of the examined solids formed at pH 3, 6 and 9 in the analogous suspensions were determined using the above-mentioned apparatus (based on the static light scattering phenomenon).

### 3.5. Adsorption–Desorption Studies

The adsorbed/desorbed amounts of PAA and PEI were determined using the static method based on the change of the adsorbate concentration in the solution before and after the process. The polymer concentrations were measured using the UV–Vis spectrophotometer Carry 100 (Varian, Palo Alto, Santa Clara, CA, USA). The PAA concentration was determined based on its reaction with hyamine 1622 resulting in formation of a white-colored complex, absorbing light at the wavelength 500 nm [31]. In turn, the PEI concentration was determined based on its reaction with CuCl_2_, which gives a blue-colored complex, absorbing light at the wavelength of 285 nm [32]. The adsorption isotherms of single adsorbates were obtained using the solutions with initial concentration changing in the range of 20–400 ppm. The suspensions were prepared by the addition of 0.01 g of activated carbon to the 10 cm^3^ of solution containing 0.001 mol/dm^3^ of the supporting electrolyte and the appropriate adsorbate. The adsorption process was carried out for 24 h at pH 3, 6 and 9. The adsorption kinetics were studied at pH 3 at specific time intervals, after 10, 30, 60, 90, 120, 180 min, using the analogous procedure of concentration determination as well as the adsorbates with the initial concentration of 100 ppm. The obtained data were fitted to the pseudo first-order (Equation (1)) and the pseudo second-order models of adsorption (Equation (2)) [33]:(1)$$\frac{dq_{t}}{dt}=k_{1}\left(q_{e}-q_{t}\right)$$
(2)$$\frac{dq_{t}}{dt}=k_{2}{\left(q_{e}-q_{t}\right)}^{2}$$
where *q_e_* isthe adsorbed amount at the equilibrium state [mg/g], *q_t_* isthe adsorbed amount after time t [mg/g], *k_1_* is the equilibrium rate constant [1/min], and *k_2_* is the equilibrium rate constant [g/(mg·min)].

The adsorption from the binary solutions was performed at the adsorbate initial concentration of 100 ppm, for 24 h. The adsorption on the biochars and the activated carbons was examined using a suspension containing 0.25 g of the biochar and 0.01g of activated carbon, 100 ppm of the appropriate adsorbate and 10 ppm of NaCl (at pH 3, 24 h). After the adsorption was complete the solids were separated from the solutions using a microcentrifuge (Centrifuge MPW 233e MPW Med. Instruments, Warsaw, Poland) and the concentration of adsorbates in the supernatants determined. The separated solids with the adsorbed polymers were, next, subjected to the desorption process (for 24 h) using the H_2_O, HNO_3_ and NaOH solutions (the acid and base with concentrations of 0.1 mol/dm^3^). All desorption tests were carried out at 25 °C.

## 4. Conclusions

The activated carbons obtained from nettle and sage herbs had largely developed surface areas. The biochars obtained from the same precursors possessed a significantly smaller surface area, which confirmed that the activation process had a huge impact on the surface area, as well as on development of the porous structure. The obtained adsorbents were characterized by a mesoporous structure which favored the adsorption of pollutants with large molecular sizes. The activation process also affected the acidic/basic properties of solids (biochars are basic in nature, while activated carbons exhibit acidic character at the surface).

It was proved that carbonaceous materials obtained from nettle and sage herbs can be used as effective polymer adsorbents from aqueous solutions. Poly(acrylic acid) was better adsorbed on the surface of all examined solids than polyethylenimine. In turn, the greatest adsorption of both polymers was observed at pH 3 on the NE_AC surface (the maximum value of the PAA adsorption was 273 mg/g, whereas for PEI it was156 mg/g). The PEI adsorption at pH 3 was not affected by poly(acrylic acid) with the coiled conformation. In turn, PEI molecules with highly developed conformation could block the porous structure of activated carbons for PAA adsorption (PAA adsorbed amount decrease from mixed system of adsorbates). It was also proved that the regeneration of the examined solids was possible; however, at a level not exceeding 50%.

## Figures and Tables

**Figure 1 molecules-27-07557-f001:**
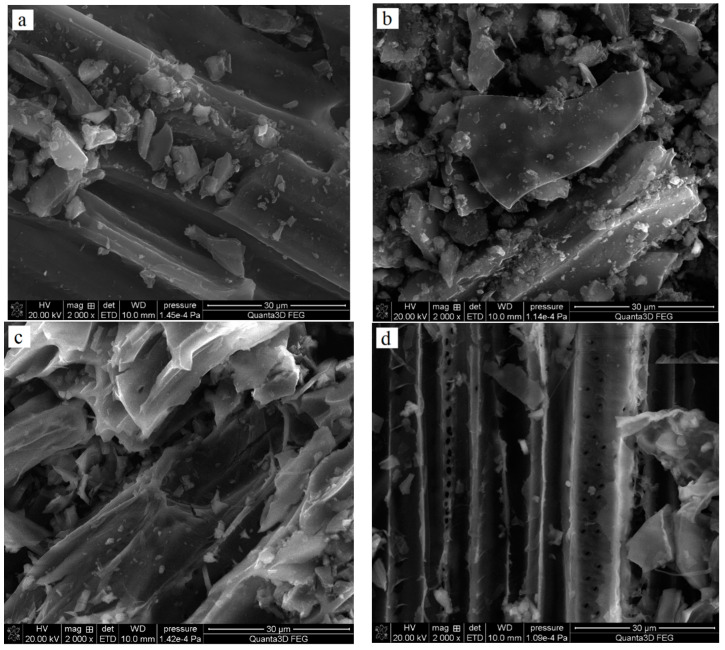
SEM images of NE_AC (**a**) and SA_AC (**b**) activated carbons and NE_B biochar (**c**) and SA_B (**d**) biochar.

**Figure 2 molecules-27-07557-f002:**
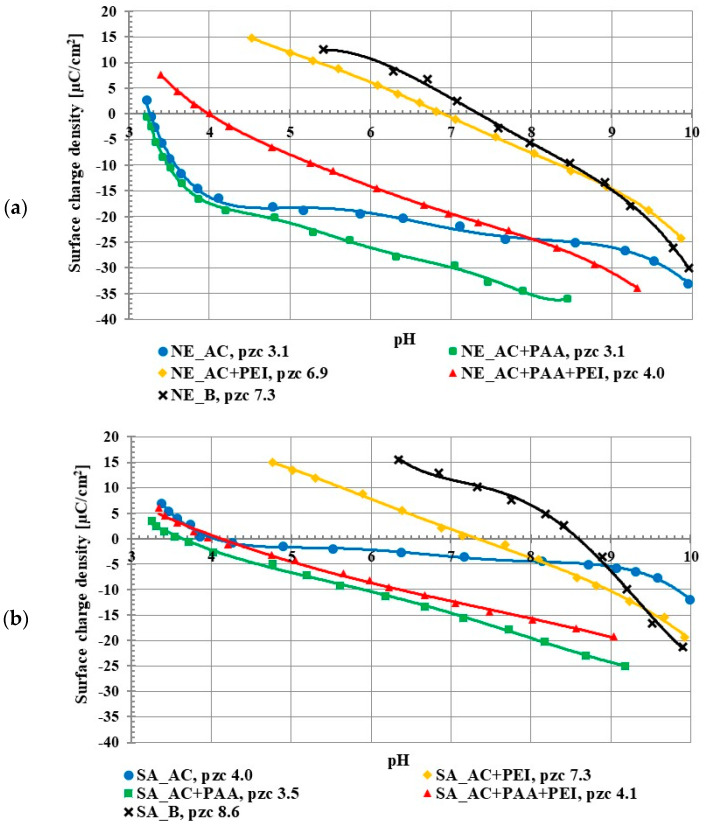
Surface charge density of NE_AC activated carbon and NE_B biochar (**a**), SA_AC activated carbon and SA_B biochar (**b**) in the systems without and with the single and binary adsorbates (C_0_ 100 ppm).

**Figure 3 molecules-27-07557-f003:**
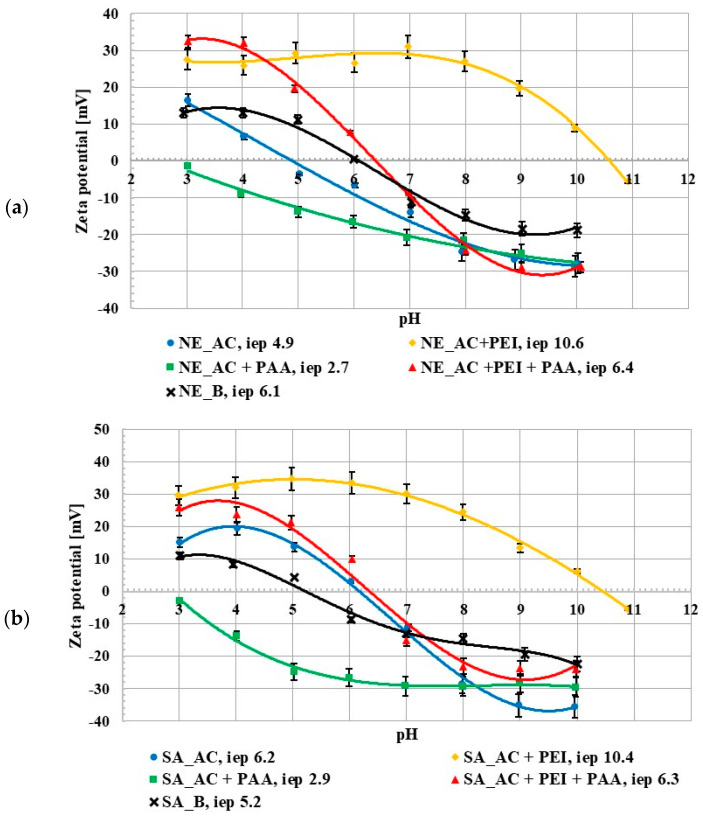
Zeta potential of NE_AC activated carbon and NE_B biochar (**a**), SA_AC activated carbon and SA_B biochar (**b**) particles in the system without and with the single and binary adsorbates (C_0_ 100 ppm).

**Figure 4 molecules-27-07557-f004:**
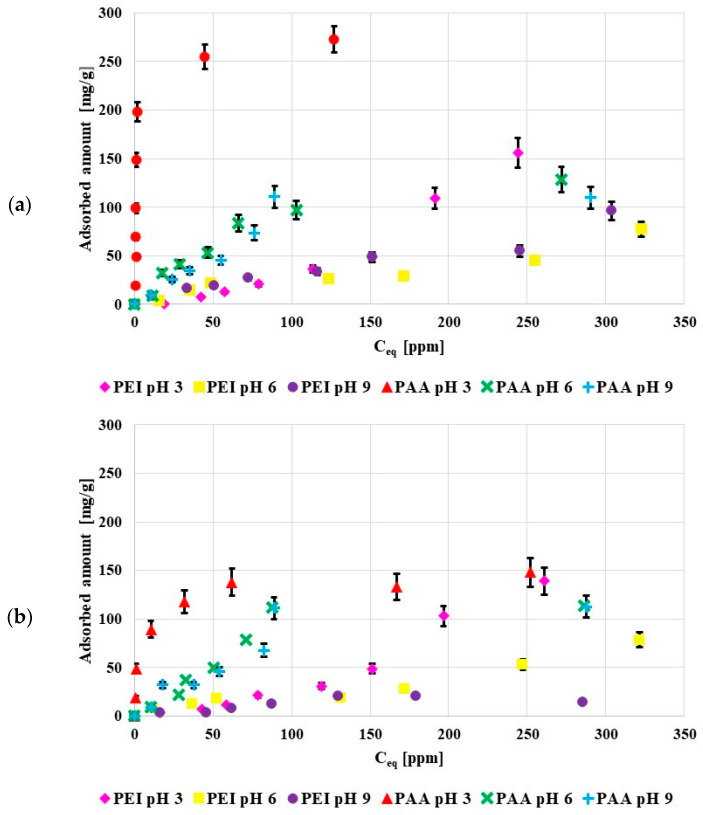
Poly(acrylic acid) and polyethyleneimine adsorption isotherms on the surface of NE_AC (**a**) and SA_AC (**b**) activated carbons (C_0_ 100 ppm).

**Figure 5 molecules-27-07557-f005:**
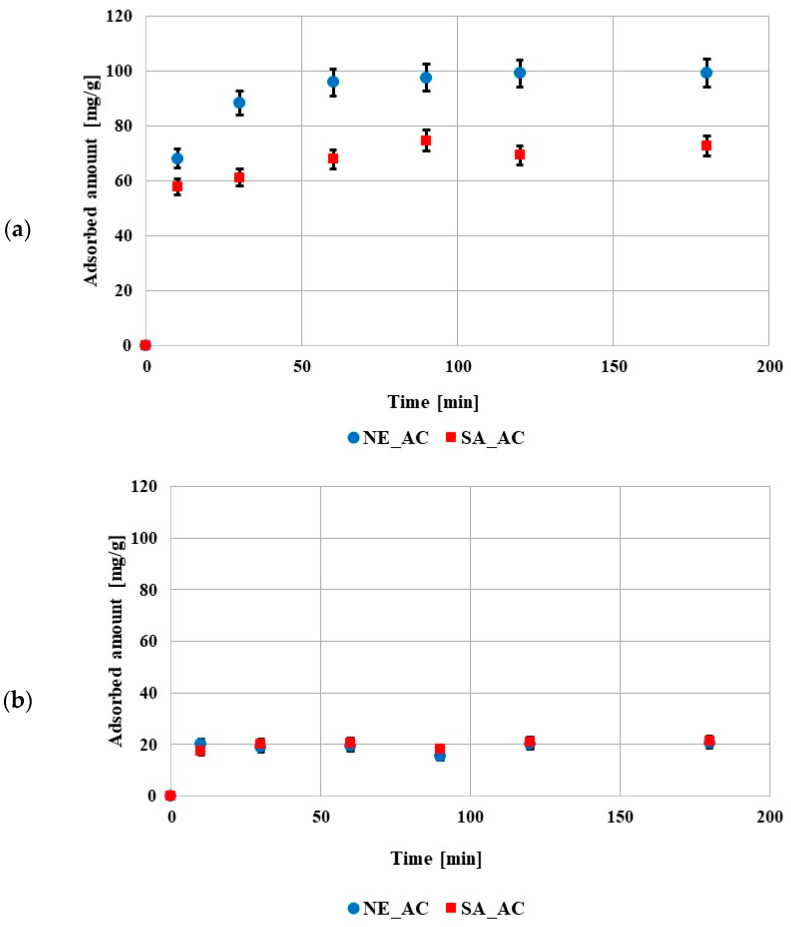
Adsorption kinetics of poly (acrylic acid) (**a**) and polyethyleneimine (**b**) on both activated carbons surfaces (pH 3, C_0_ 100 ppm).

**Figure 6 molecules-27-07557-f006:**
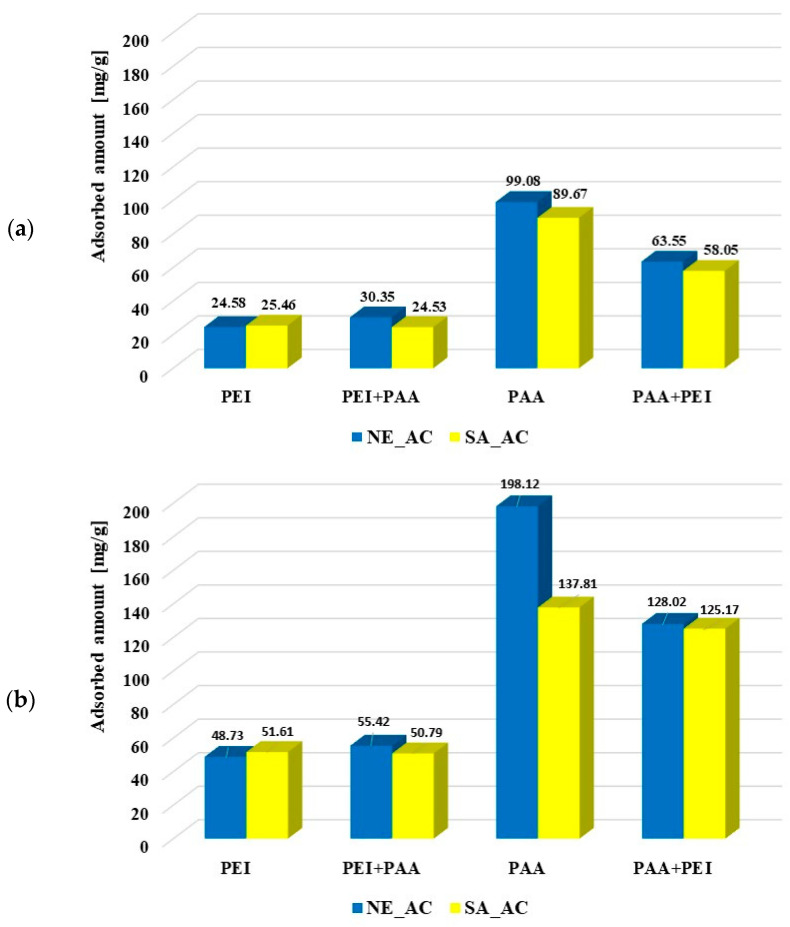
Poly (acrylic acid) and polyethyleneimine adsorbed amounts on the activated carbons surface at pH 3 from the single and binary solutions of the initial concentrations C_0_ 100 ppm (**a**) and C_0_ 200 ppm (**b**) as well as on the biochars’ surfaces (C_0_ 100 ppm) (**c**).

**Table 1 molecules-27-07557-t001:** Elemental composition of the obtained adsorbents [wt. %].

Adsorbent	C	O	N	P	Ca	Mg	K	Others
NE_AC	90.62	7.66	1.13	0.59	-	-	-	-
SA_AC	89.61	8.81	0.24	0.63	-	-	-	0.71
NE_B	78.76	16.35	2.77	0.14	1.23	0.27	0.30	0.18
SA_B	81.31	15.50	1.78	0.20	0.47	0.38	0.15	0.21

**Table 2 molecules-27-07557-t002:** Textural parameters of the prepared activated carbons and biochars.

Adsorbent	Pyrolysis/ Activation Yield [wt. %]	Surface Area [m^2^/g]		Pore Volume [cm^3^/g]		Mean Pore Size [nm]	Micropore Contribution
		Total	Micropore	Total	Micropore		
NE_AC	37.7	801	157	0.847	0.074	4.231	0.087
SA_AC	40.3	842	155	0.826	0.074	3.926	0.090
NE_B	42.9	2.5	-	0.006	-	9.594	-
SA_B	45.8	2.1	-	0.006	-	10.555	-

**Table 3 molecules-27-07557-t003:** Acidic-basic properties of the prepared biochars and activated carbons.

Adsorbent	Acidic Groups [mmol/g]	Basic Groups [mmol/g]	Total Amount [mmol/g]
NE_AC	0.858	0.272	1.130
SA_AC	0.436	0.215	0.651
NE_B	1.041	1.753	2.794
SA_B	1.109	1.008	2.197

**Table 4 molecules-27-07557-t004:** Aggregates sizes of activated carbons and biochars in the system without the polymers as well as in the single and binary adsorbates.

System	Size of Aggregates [nm]		
	pH 3	pH 6	pH 9
NE_AC	509.5	478.2	448.5
NE_AC + PAA	499.2	405.4	433.7
NE_AC + PEI	459.1	516.4	519.3
NE_AC + PAA + PEI	599.9	1099.7	741.9
SA_AC	327.8	430.9	517.4
SA_AC + PAA	701.0	441.4	4002.0
SA_AC + PEI	475.0	842.1	370.9
SA_AC + PAA + PEI	204.0	874.6	289.2
NE_B	1257.0	731.2	1448.0
SA_B	688.2	1799	1494.0

**Table 5 molecules-27-07557-t005:** Kinetic parameters of the PAA/PEI adsorption on the activated carbons.

Calculated Parameters	Pseudo-First-Order Model			Pseudo-Second-Order Model		
	q_e_ [mg/g]	k_1_ [1/min]	R^2^	q_e_ [mg/g]	k_2_ [g/(mg·min)]	R^2^
	**PAA**					
NE_AC	1.03624	8.03309	0.9376	102.041	0.00237	0.9999
SA_AC	1.02768	6.86179	0.9556	71.9424	0.00476	0.9976
	**PEI**					
NE_AC	1.0329	1.22266	0.6614	20.5339	0.05842	0.9693
SA_AC	1.01837	1.6098	0.699	21.322	0.05418	0.9913

**Table 6 molecules-27-07557-t006:** Content of elements in the surface layer of NE_AC carbon without, and in the presence of, polymeric adsorbates.

Element	NE_AC	NE_AC + PAA	NE_AC + PEI	AC + PAA + PEI
**Cl** [at. %]	-	-	1.3	0.8
**N** [at. %]	1.1	0.6	2.9	5.4
**O** [at. %]	8.6	16.6	13.0	16.1
**P** [at. %]	0.8	0.6	0.6	0.8
**C** [at. %]	89.5	82.2	82.2	76.9

**Table 7 molecules-27-07557-t007:** Percentage desorption of poly(acrylic acid) and polyethyleneimine from the activated carbons and biochar surfaces from the single and binary systems using H_2_O, HNO_3_ and NaOH.

Desorption Agent	Desorption [%]		
	H_2_O	HNO_3_	NaOH
System	PAA		
NE_AC + PAA	1.36	2.76	61.01
NE_AC + PAA + PEI	1.23	1.81	47.68
SA_AC + PAA	2.53	2.00	46.45
SA_AC + PAA + PEI	1.36	2.42	48.59
NE_B + PAA	6.57	8.95	-
NE_B + PAA + PEI	34.19	7.08	-
SA_B + PAA	18.01	6.40	-
SA_B + PAA + PEI	2.19	7.81	-
**System**	**PEI**		
NE_AC + PEI	44.15	-	-
NE_AC + PEI + PAA	4.52	-	-
SA_AC + PEI	24.73	-	-
SA_AC + PEI + PAA	9.69	-	-
NE_B + PEI	-	-	-
NE_B + PEI + PAA	-	-	-
SA_B + PEI	44.08	-	-
SA_B + PEI + PAA	34.80	-	-

## Data Availability

Data are contained within the article.

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
