# Peer review of "Simultaneous Removal of Polymers with Different Ionic Character from Their Mixed Solutions Using Herb-Based Biochars and Activated Carbons"

_molecules, 2022, doi:10.3390/molecules27217557_

Round 1

Reviewer 1 Report

molecules-1996988                                                                                    Research Paper 

Title: Simultaneous removal of polymers with different ionic character from their mixed solutions using herb-based biochars and activated carbons

Authors: Gęca et al.

General]

The authors synthesized biochars and activated carbons for the adsorptive removal of polymers from aqueous solutions. The manuscript can be considered for publication, subject to significant corrections.

Specifics]

1] The manuscript requires professional English proofreading due to incorrect grammar and strange lexical choices at several spots.

2] All experiments should be repeated to ensure data reproducibility. Vertical error bars should be provided at each data point.

3] As real-world wastewater systems are a complex mixture of organic and inorganic compounds, how will the co-presence of competing species affect the polymer adsorption performance of the tested adsorbents?

4] X-ray photoelectron spectroscopy (XPS) data of the adsorbents should be provided before and after polymer adsorption to propose a suitable mechanism.

5] Why did the adsorbents prefer adsorbing one of the polymers more favorably over the other? Suitable explanations should be provided.   

Author Response

Reviewer 1

  1. The manuscript requires professional English proofreading due to incorrect grammar and strange lexical choices at several spots.

The English was checked and corrected.

  1. All experiments should be repeated to ensure data reproducibility. Vertical error bars should be provided at each data point.

The error bars were added to the electrokinetic and adsorption data.

  1. As real-world wastewater systems are a complex mixture of organic and inorganic compounds, how will the co-presence of competing species affect the polymer adsorption performance of the tested adsorbents?

We agree with Reviewer that real wastewater has very complex composition. For this reason we investigate the simultaneous adsorption of two polymeric substances with different ionic character. Currently, our research is conducted in an even more complex system of adsorbates, i.e. in addition to polymers, they include heavy metals – cadmium and arsenic. These metals differ in ionic forms in which they occur in aqueous solution. We also plan to examine the influence of surfactants. Such complex systems containing more than two adsorbates are extremely rare investigated, that’s why our studies could be very helpful in understanding of adsorption mechanism in multicomponent solutions.

  1. X-ray photoelectron spectroscopy (XPS) data of the adsorbents should be provided before and after polymer adsorption to propose a suitable mechanism.

The XPS data were presented (Table 6) and discussed in the manuscript.

  1. Why did the adsorbents prefer adsorbing one of the polymers more favorably over the other? Suitable explanations should be provided.

Obtained results indicated that poly(acrylic acid) is adsorbed better on the activated carbons than polyethylenimine. Most probably the polymers conformation assumed at pH 3 (PAA – coiled, PEI – developed) is responsible for that. Both activated carbons have similar textural properties so both types of macromolecules are adsorbed on their surface at a similar level. The polyethyleneimine adsorption is not affected by poly(acrylic acid). Small PAA molecules (with the coiled conformation) are probably adsorbed in the pores, so they do not impede the PEI adsorption onto the solid surfaces. However, the developed PEI molecules can block the porous structure of activated carbons for the poly(acrylic acid) adsorption, which resulting in the PAA adsorbed amount decrease, in comparison to its adsorption from the single solution. The polymer-polymer complexes formation can also affect the macromolecules adsorption from the binary solution, however, at pH 3 these complexes are hardly formed due to the coiled conformation of undissociated PAA chains.

The above explanation was presented in the text.

Reviewer 2 Report

The author reported nettle and the sage herbs derived biochar and activated carbons, which exhibited poly(acrylic acid) and polyethylenimine adsorption. The textural parameters of the catalysts are well characterized. The adsorption parameters and mechanism are discussed in detail. I recommend this manuscript for publication in molecules, however, there are several issues that need to be addressed as follows.

(1) The abbreviations and full names of PAA and PEI appeared several times in abstract.

(2) The full names of four catalysts should be mentioned before abbreviations in the manuscript.

(3) Page2, Line 89, what does Nadf mean.

(4) Several mistakes existed in this manuscript, such as Page 2 Line 89.

(5) what is the difference between point of zero charge and isoelectric point?

(6) In table 2, the SBET of NEAC and SAAC are above 800, while that of NEB and SAB are only 2.5 and 2.1. However, the total amount of acidic-basic sites is higher in NEB and SAB than NEAC and SAAC. The author should explain these phenomena.

(7) Again, the SBET of NEAC and SAAC are above 800, while that of NEB and SAB are only 2.5 and 2.1. However, the size of aggregates of NEB and SAB are only slightly higher than that of NEAC and SAAC. The author should explain these phenomena.

Author Response

Reviewer 2

  1. The abbreviations and full names of PAA and PEI appeared several times in abstract.

Polymer abbreviations are given in the abstract together with their full names when they are firstly mentioned. In the following fragments they were used interchangeably, either the full name of the polymer or its abbreviation.

  1. The full names of four catalysts should be mentioned before abbreviations in the manuscript.

The full names of applied biochars (NE_B, SA_B) and activated carbons (NE_AC, SA_AC) obtained from the nettle (NE) and the sage (SA) herbs were provided in Materials and Methods section (subsection 3.2. Biochars and activated carbons preparation). Their abbreviations were used in Results and Conclusions sections.

  1. Page2, Line 89, what does Ndaf

Parameter Ndaf (daf - dry ash free) results from the conversion of the elemental composition into ash-free dry substance. It was explained in the text.

  1. Several mistakes existed in this manuscript, such as Page 2 Line 89.

The English was checked in the whole manuscript. The grammatical errors were corrected.

  1. What is the difference between point of zero charge and isoelectric point?

The point of zero charge (pzc) is a pH value at which the concentration of positively charged surface groups is the same as the concentration of negatively charged ones, so the total surface charge is equal to zero. In turn, the isoelectric point (iep) is a pH value at which the number of adsorbate ions/functional groups with a positive and negative charges, accumulated in the slipping plane area, are the same (the zeta potential has a zero value).

Thus, the pzc refers to the solid surface layer, whereas the iep to the slipping plane area. Both of them are parts of electrical double layer (edl) formed at the solid/liquid interface.

Such explanation was added in the text.

  1. In table 2, the SBET of NEAC and SAAC are above 800, while that of NEB and SAB are only 2.5 and 2.1. However, the total amount of acidic-basic sites is higher in NEB and SAB than NEAC and SAAC. The author should explain these phenomena.

Studies carried out by many researchers indicate that there is no direct relationship between the specific surface area of carbon materials and the amount of acidic or basic functional groups. In case of biochars and activated biocarbons, the total amount of surface groups determined using titration method is also influenced by the alkaline mineral substance present in their porous structure. Moreover, the pyrolysis temperature of biochar (400 oC) did not guarantee the removal of volatile organic compounds. This process was more effective at temperature of 500 oC, in which activated biocarbons were obtained. Additionally, the use of phosphoric(V) acid as an activating agent resulted in a partial hydrolysis of lignin, cellulose or hemicellulose present in the precursors structure. All these effects influence the total amount of acidic-basic groups. It should be also noted that high concentration of surface groups can cause blockade of porous structure of carbon materials making it inaccessible for the adsorption of nitrogen molecules (used during determination of the specific surface area).

  1. Again, the SBET of NEAC and SAAC are above 800, while that of NEB and SAB are only 2.5 and 2.1. However, the size of aggregates of NEB and SAB are only slightly higher than that of NEAC and SAAC. The author should explain these phenomena.

The size of formed aggregates is not directly related to the development of specific surface area of solid particles. Aggregation tendency depends primarily on the interaction between the particles of carbon materials and it is limited to the nature of the external surface of the adsorbent (the nature of the pore surface is irrelevant regarding this). Polymer adsorption layers change the nature of these interactions (which are mainly based on electrostatic forces), because in such a case there is also a steric component resulting from the presence of spatial obstacle formed by adsorbed macromolecules with a specific conformation.

Round 2

Reviewer 1 Report

The authors have suitably addressed my comments. The manuscript can be accepted for publication.

Reviewer 2 Report

The referee has no more question